# Meaning Analysis and Alienation: A Method of Immanent Critique in Acute Psychiatry

**DOI:** 10.3390/ijerph192316194

**Published:** 2022-12-03

**Authors:** Frieder Dechent, Julian Moeller, Christian G. Huber

**Affiliations:** 1University Psychiatric Clinics Basel, University of Basel, 4002 Basel, Switzerland; 2Division of Clinical Psychology and Epidemiology, Department of Psychology, University of Basel, 4055 Basel, Switzerland

**Keywords:** acute psychiatry, immanent critique, meaning, Niklas Luhmann, Rahel Jaeggi, alienation

## Abstract

In acute psychiatry, where people with severe mental disorders are frequently treated, there can be contradictions between concepts of illness among, e.g., patients and healthcare professionals, and also between medical and legal aspects. These contradictions do not manifest themselves openly but are immanent in the social practices of the treatment teams as contradictions between the social level and the individual level. They can lead to alienation, which may be reflected in poorer quality of treatment, such as the more frequent use of coercive measures or poorer adherence to therapy in patients. In the normal daily routine of a clinic, these contradictions are mostly hidden by hierarchical structures or by unbalanced concepts of psychiatric illness, or external critique is used to try to solve these contradictions. However, another way of dealing with these contradictions could be to analyze the potential and causes for alienation through systematic analysis and transformation of the whole system of a psychiatric ward to reduce the level of contradiction within it. The aim of this work is to use the concept of meaning elaborated by Luhmann to identify and recognize alienation potentials as concretely as possible and thus make them accessible to immanent critique. Meaning in Luhmann’s use of the term serves to reduce complexity in a social context and always opens up consequential possibilities for action. Consequential limited possibilities at the level of action in a rigid social system (which psychiatric wards can be) can—at an individual level—lead to people subordinating themselves to the rigid system to an excessive degree and thus alienating themselves from the system. Thus, a rigid system with a narrowing of consequential possibilities excludes meaningful consequential possibilities. This leads to alienating contradictions and to possibilities of world appropriation being missed. The aim of the current analysis is not to make a general critique of psychiatry but to improve the theoretical basis to better understand the problem of alienation in acute psychiatry as a symptom of system-immanent contradictions and thus open up the possibility of transforming systems, e.g., psychiatric acute care units, by means of immanent critique.

## 1. Introduction

The United Nations Convention on the Rights of Persons with Disabilities, 2007 [1] strengthens the rights of persons with mental disorders and emphasizes their rights of self-determination, autonomy, and participation in society. Despite the legal requirements, implementation and interpretation at different levels and in different institutions appear to vary significantly.

Particularly in the field of acute psychiatry, which is often mandated with the treatment of involuntarily admitted patients, systematic contradictions appear to be quite commonplace, especially with regard to legal requirements and different human and disease profiles at the individual, ward and clinic levels. According to the current study, such contradictions can cause alienation among both patients and healthcare professionals, which in turn can lead to poorer treatment outcomes.

The question arises how system-immanent conflicts can be analyzed and transformed to reduce alienation in the social system of a psychiatric ward among patients and healthcare professionals, both at the individual and social levels. Thus, it is not a matter of criticizing acute psychiatry from external or internal perspectives—and thereby implementing a normative idea—but rather of being able to exercise a theory-guided immanent critique, whereby “*the new emerges as an absorbing transformation of the old*” [2].

In the current literature, there are numerous examples that point out the problems at social and individual levels for staff as well as patients, especially in the inpatient treatment of psychiatric patients. In addition to actual medical problems, an essential aspect is the role in which staff find themselves, which may be in contrast to personal ideals regarding treatment [3,4]. Other problems that have been described relate primarily to poor communication and imbalances of power, which make it difficult to take the patient’s needs into account. Coercion can also be justified on the assumption that a patient cannot understand the information that is being given to them at all [5]. With regard to these problems, it seems important from our point of view not only to criticize the underlying problems from the outside but also to point out the contradictions immanent in the system, especially between the individual and social levels.

In the current study, we discuss the principle of immanent critique and introduce the concept of alienation based on Luhmann’s theory of meaning. The focus of such an analysis should be on the psychiatric ward as the environment of the patient and the staff; our theory can also offer connection possibilities from the analysis of the subsystem of the psychiatric ward to the wider environment of society as the overarching environment. The theory is ultimately intended to offer a perspective, with the help of practical references, on how, through immanent critique, systematic contradictions can be resolved or mitigated both at the individual and social levels in the treatment of patients—that is, at the ward or clinic levels—and transformation processes can be initiated.

The hypothesis that we want to examine theoretically in this work is that there are contradictions between the intentions and abilities of the individual actors in a psychiatric ward and the patterns of action that are shaped in the social system of a psychiatric ward by expectations that lead to the alienation of individuals and which make such systems alienating.

## 2. Immanent Critique

First of all, we would like to describe the theory of immanent critique so that the discussion that follows can be properly contextualized. The essential point is that it is not the object of this work to create transformation possibilities by introducing new contentual concepts; rather, the point is that an effective transformation also includes the clarification of immanent contradictions in the social system of a psychiatric ward and the people involved. Another central point is that, in this text, we should focus on the contradictory relationships and attributions between the actors and the resulting alienating system.

Tensions and crises in a social system can be clarified and, at least partially, resolved by immanently criticizing the system—in our case, the system of an acute psychiatric ward—which means that the system is confronted with its own values and norms, without applying external values and norms as measures [6]. Immanent critique comes into play in this form when there are conflicts and contradictions immanent in reality, which lie in the matter itself [2]. Titus Stahl describes these contradictions between explicit norms, that is, norms that every member of a community agrees to, and immanent norms, which are implicitly anchored in social practice, in detail, always emphasizing the corresponding need for explanation:


*“The objectively diagnosable resistance to ‘official’ evaluations points to the fact that other norms have emerged on the level of intersubjective recognition, which are not able to be articulated on the level of the ‘official’ norms. […] When a group in society systematically accepts norms in an inarticulate way that contradicts official social norms, there is a need for explanation.”*
[7]

According to Stahl, the reason for the emergence of such contradictions is that these implicit norms are not openly formulated. The acceptance of criticism is, therefore, also fundamentally related to possible reasons, such as: “*the systematic exclusion of certain positions and norms from official discourses, deficiencies in conceptual competence and discursive self-awareness due to educational deficits, structural pathologies of discourse situations, taboos, and many more*” [7].

In this context, Jaeggi refers to the functional–constitutive character of norms, stating a systematic necessity for norms in a society and that social norms constitute social reality [2].


*“It is thus not merely the case that in a given social context certain normative principles are (more or less accidentally) espoused which are then failed in the same context; but the corresponding social practices and institutions are themselves constituted by those norms which they simultaneously contradict.”*
[2]

In immanent critique, no new norm is set in opposition to the existing one; rather, the existing relations are transformed, whereby “*the new emerges as an absorbing transformation of the old*” [2]. Another characteristic of immanent critique, according to Jaeggi, is that it is not fixed, but transforms itself during the process of critique and must, therefore, always justify itself during the process of critique [2]. Criticism is also not subsequent to an analysis: it “*is critique as analysis […] and analysis as critique*” [2].

Immanent critique is thus not to be understood as a method that serves a superordinate norm and attempts to measure—and subsequently align—real processes and events against this norm but as one that rather attempts to recognize and transform in the process itself the contradictions between practice and norms, which in this sense are “*constitutive for the functioning of the society thus described and […] decisive for the self-understanding of those participating in this formation*” [2]. A circular process emerges, resulting from an alternation between critique and feedback from application, so that both sides can transform.

Immanent critique needs to be examined closely on several points. Martin Weißmann [8], for example, cites two problematic premises of immanent critique: first, the claim that immanent critique is not normativistic, that is, that it applies only those value standards that come from within the social system to be criticized, and, second, “*that a social formation identified as systematically normative in itself is deficient, problematic, and unstable*” [8]. Regarding the first premise, he states that even immanent critics are bound by external norms when making directional decisions, which is certainly partially true for institutions such as psychiatry, because its borders open into different fields. Existing norms cannot change in arbitrary directions either, but will come into contradiction with other norms and practices that may be at issue; some of the practices also violate, or involve an interpretation of, the norms that can, in principle, be regarded as open to criticism. However, this also shows that external norms, such as laws, practically always have a constitutive character in institutions, although their normative force is always questionable and open to interpretation. In our opinion, however, there is no need for additional external norms, because directional decisions should often be derived from public discourse in the system, and it is not for the external critic to indicate contradictions to help make this directional decision. What is essential at this point for our procedure is that the relevant legislation, guidelines or even ethical principles can always be brought into the system from outside. However, as it is not a closed system, these can be accepted in their superficial necessity, in part, but in principle, through their integration process, they become immanent in the system and thus, in turn, also become part of a transformation process.

However, even such inputs from outside the system can only be effective within the system if the basic requirements for transformability are given. For example, new therapies, ward concepts or even individual ethical attitudes can only be effective if the systemic social structure, which we will examine more closely in the next section, does not become closed off to criticism.

Weißmann [8] notes that the ideal of freedom from contradiction cannot necessarily be achieved and that contradictions in a system also partly improve the system’s ability to learn and reflect. These objections should be kept in mind, because the goal of complete freedom from contradiction cannot be achieved in a complex and dynamic entity with partly predetermined framework conditions, and certain norms can only be partly transformed in terms of their interpretation. Finally, crises should provoke criticism and not a permanent struggle for freedom from contradictions. Moreover, in a critical area, such as acute psychiatry, the absence of contradictions seems difficult to achieve because patients often move in a context that is inevitably characterized by contradictions in their own ideas and needs. In this context, it should be the task of immanent critique to confront these system-immanent contradictions and, even if total resolution is not possible, to negate the crisis potential of these contradictions in an ongoing process.

In summary, our text is about not only relating the immanent critique to the contradictions between overarching topics, such as law, ethics or patient autonomy, and the norms and practices of a psychiatric ward, but rather about pointing out contradictions between the individual and social levels. In the following, we will present a possibility of making this structural problem of the social system accessible to an immanent critique and thus to transformation. We will make it clear that inpatient treatments in particular, as social practices, are also inherently open to criticism, because they are structured by expectations and the expectation of expectations. The resulting social system can oppose the wish of every actor, patient or staff, to act according to norms or their own therapeutic possibilities and can therefore be alienating.

One important remark should be made. Based on our experience, it can be assumed that the patient who is admitted against his or her will also finds himself/herself in unavoidable and alienating contradictions with his/her social reference system outside of the clinic, which leads either to actual danger or to a risk that is assumed with great probability, which usually justifies hospital admission from the point of view of the admitting person. In this paper, we mainly examine the immanent contradictions in the psychiatric ward; for a wider analysis, the scope could also be extended. From the perspective of the patient, the contradictions from outside continue in the clinic. Even if the focus of this work is not primarily on this topic, from the perspective of our theory, the treatment goal can be seen in the fact that the patient is given assistance through the best possible solution or mitigation of contradictions in the clinic rather than contradictions that lie outside, which contradictions are to be confronted within the clinic so as to enable the patient to deal with them better.

## 3. Case Study

The following constructed case study, though intended to be explanatory, is deliberately simplified, as opposed to an actual empirical analysis, to allow for methodological analysis within the context of the current study. In order to keep the study short and also to preserve its illustrative character in the overall context, we would like to limit ourselves to a single central problem which can occur again and again in various forms in situations in acute psychiatry and which, by way of example, illustrates the possibilities that should be applied to criticism. The focus of our exemplary presentation should be psychopharmacotherapy, in particular, the distribution of roles in the decision-making process. The limitation of this representation is certainly that the full complexity cannot be fully represented within the framework of this case study; however, it can still serve to represent a fundamental social mechanism that can have an alienating effect in these decision-making processes. Another limitation of this type of presentation is that the presentation can be perceived as overly critical; a real case study would certainly offer more space for a more differentiated presentation of positive aspects.

The method as well as the case study were developed from our own clinical experiences against the background of the repeatedly arising question of why social subsystems can exhibit a rigidity that prevents an adequate implementation of the existing therapeutic resources and why their functioning as a whole leads to alienating contradictions between individual and system levels without these contradictions being open to clear criticism. It should also be mentioned at this point that, in our clinical experience, a reduction in rigidity also results in the improved transformability of the social system in terms of improved use of available resources for the patient and thus reduced alienation among both staff and patients. In connection with the methodology presented here for criticism, our case discussion contains abbreviations and also exaggerations in many respects; hence, it should not serve as an example of poor treatment in terms of content, in the sense of a deviation from guidelines in a real case; rather, our concern is to describe and criticize social “pathology” as an alienating and alienated system. In particular, we would like to point out that the case study is not intended to serve as a general criticism of psychiatry but rather to explain in an exaggerated way possible mechanisms that can make the best possible treatment more difficult. In our view, psychiatric acute care units, as social subsystems, like most other social subsystems are on a continuum between well-functioning and poorly functioning systems that are alienating for the individuals concerned. The problems described in our case study are therefore often non-existent or only slightly present in reality. We therefore consider the extreme case of an alienating crisis as an example.

Police bring a young patient to a clinic against his will because of conspicuous behavior in public spaces, in particular, harassing passers-by. The doctor who was responsible for the involuntary admission to the clinic admitted the patient because of an acute danger to others. In the admission interview, he expresses dissatisfaction with the admission, denies the circumstances of admission and refuses psychiatric treatment. Previous history reveals a diagnosis of paranoid schizophrenia with acute episodes requiring hospitalization, which is why the patient was repeatedly in the clinic and either refused antipsychotic medication or discontinued it shortly after release, once there was improvement. The psychopathological findings show a pronounced experience of persecution and impairment, sensory illusions in the form of pejorative voices and formal mental incoherence. He appears slightly tense—which means that he does not pose an acute danger to himself or others—which would justify further coercive measures. The patient rejects antipsychotic pharmacotherapy. The next day, he appears unchanged, and the psychopathological abnormalities persist. Due to the psychopathology, the ward team repeatedly try to convince the patient to undergo pharmacological therapy; the attempts, however, do not succeed. The patient was always suspicious in contact and in the psychotherapeutic sessions and the sessions with the nursing staff, because the necessity of medication was often discussed there, too, and the patient did not feel that this topic was being taken seriously.

The medical psychoeducational talks served, above all, to gain a biological understanding of the disease.

Attempts are made to implement other interventions and social planning strategies, but only to a limited extent given the fundamental necessity of psychopharmacological treatment from the team’s perspective. After the patient has been in a closed ward for a long duration, he does not return from one of the first excursions.

In the final supervision, there was great dissatisfaction among the team due to the course of treatment and the options that had not been exhausted. It was emphasized by all participants that the treatment resources available to the team were not used. Both the psychotherapeutic side and the medical side noted that there was no room for specific psychotherapeutic methods that were aimed specifically at the psychotic symptoms independently of psychopharmacological treatment and which could have opened up access to the patient, since the focus was primarily on pharmacology, although the therapeutic resources were available to the treatment team. The nursing side noted that the patient’s needs could not be heard sufficiently, and since the medication was offered and rejected by the patient, no relationship that the patient considered helpful could be established, although their goal was to support the recovery approach. Ultimately, it was not possible to say exactly why the treatment took this course.

In this case study, various actors with different goals, requirements and attitudes can be distinguished from one another. The focus is on the patient who wishes to be discharged from the clinic immediately and has no insight into his psychiatric disorder and therefore rejects the treatment offered. He distrusts the treatment team due to the circumstances of admission and his experience of previous stays and feels pressured into taking medication, which he refuses, and therefore he cannot accept the treatment efforts of the team. The doctor in charge of the case sees himself very much in his role as being responsible for the pharmacology in the context of the team and therefore repeatedly emphasizes the biological understanding of the disease as appropriate to his role. Although he is a trained psychotherapist, he regards other treatment options as being more the responsibility of the other team members, given his understanding of roles, which has grown over time. The psychologist sees the main goal of successful treatment as relationship building in order to subsequently develop an understanding of the disorder with the patient and, if this is not possible, to be able to achieve successful treatment without medication using psychotherapeutic interventions. However, he is consistently confronted with his social role, which requires him to convince the patient of a course of psychopharmacotherapy, and due to the patient’s distrust based on his experiences regarding the expectations of undergoing drug therapy, building a relationship is made significantly more difficult. Overall, the nursing team is very heterogeneous, due to different personal attitudes and professional experiences. Some certainly also see drug treatment as the priority, while others are concerned about the treatment direction taken by the doctor and also see ways to help the patient within the framework of their personal abilities and professional skills but feel that their competence is not recognized or that there are no possibilities for applying them. Nevertheless, basically, they all agree about the recovery concept actually implemented on the ward. Other professional groups, such as ergotherapists and social workers, are not included for the sake of simplicity, but usually they will play a role in the overall dynamic as well. (At this point, it is important for us to emphasize that the fact that the doctor is drawn as a representative of psychopharmacology is only one variation. For example, the physician could be involuntarily cast in the role of representative of psychopharmacology. He himself would also have other therapeutic options but may feel compelled to act in accordance with the role due to entrenched job-specific expectations, both his own and those of the other employees. Similarly, the caregivers may be particularly strong advocates of psychopharmacology, e.g., due to the expectations that treatment will otherwise fail and that behavior cannot be controlled. Unfortunately, however, it is not possible within the scope of the present article to discuss these variants in detail or to work through several theoretical cases. That would, rather, be the subject of empirical research).

At first glance, we have a typical clinical department with a clear allocation of roles. Each team member sees themself in a role, and, to put it simply, this role is also continuously confirmed by the rest of the team. On the one hand, this is achieved through the clear distribution of competences and hierarchy; on the other hand, this definition can also be regarded as fixed before the time horizon. The patient does not feel adequately perceived in this system. In this form, this system is alienated as well as alienating, since the actors in this system cannot adequately relate to their possibilities. The emergent social level clearly contradicts individual motives and intentions but also overarching goals and approaches, such as the recovery model.

Of course, a psychiatric ward is also subject to numerous external influences that have a more or less direct influence on treatment. In most cases, the laws of the relevant country form the framework conditions which, under certain conditions, allow the patient to stay but also to be given medication against his or her will. Other external influences can be economic and local social influences, which at least indirectly affect the interpretation of the legal situation. Another central factor is the implementation of the UN Disability Rights Convention. In practice, these various influences often contradict each other to a certain extent, but this is not the central theme here.

What we go on to analyze based on the case study are the existing contradictions between the individual actors themselves and between the individual and the social level. In particular, it is to be shown that the social level can essentially cover up these contradictions.

## 4. Meaning as the Basis of Analysis and Criticism

In order to be able to better grasp the concept of alienation in practice and to better uncover existing contradictions, we will apply Luhmann’s theory of meaning to our case in what follows and then establish a theoretical link to alienation and immanent critique.

In order to be able to better understand Luhmann’s concept, the starting point must be to understand the social system as a complex system in which there is no basic certainty of state. For each person, the perception of another person is always linked to an experience of contingency, which means that the other person has acted, is acting or will act in one way or another based on the current situation [9,10]. (For further understanding, let us note here the difference in Luhmann’s view between psychic systems and persons, which is why, in this paper, we usually refer to persons rather than psychic systems. “We would like to call psychic systems that are observed by other psychic systems or by social systems persons” [9]. We will also not adopt a system-theoretical perspective in this paper but use Luhmann’s conception of meaning to extend the possibilities of a critical theory.) The behavior of the other person is possible but not necessary given the situation. Since the experience of contingency exists for both interaction partners, Luhmann also speaks, referring to Parsons and Shils, of double contingency [9]. Actions are not automatically mutually dependent but require communication, that is, to put it simply, a reduction in complexity, which is essentially based on generalizations and conventions [11,12]. Social systems consist of communication. Expectations and expectational expectations, i.e., expectations of what others expect from you, also serve as the basis for behavior and can thus stabilize the system [9]. One chooses one’s meaningful action because, based on one’s knowledge, one assumes that this action decision is relevant for other people or one considers the actions of others to be relevant for oneself [13].

What does Luhmann mean by the term “meaning”? The individual psychic system is forced to select in view of the complexity of the world: “*only in the world* can one learn to establish oneself as a system by selecting among possible structures” [14]. Meaning means that the respective social system or the individual person tends to prefer certain options, such that the complexity of the world is always present again and forces selection. The problem of double contingency, or its solution process, occurs due to the initial symmetrical relationship of two or more persons in a state of uncertainty via an asymmetrization to a resymmetrization in the sense of dissent or consensus [9]. If someone takes the initiative in starting to communicate, this leads to asymmetrization, which is also guided by expectations or expectations of expectations. Initiatives that hold out the prospect of prompt resymmetrization by consensus appear more promising in this context [15].

Meaning, thus, leads to the fact that the social system or individual in question tends to prefer certain possibilities, whereby the complexity of the world is again present in each case and again forces selection [16]. The same applies to social systems, which also use meaning as a form of order.

Our case study shows a situation that certainly raises some questions in the context just mentioned. On the one hand, the patient finds himself in a situation in which expectations and expectations of expectations are essential aspects, which are shaped from the outset by the psychopathology, the models of mental disorder and treatment expectations on the part of the therapists. The personal and professional experiences of the individual team members also have a significant influence. On the other hand, at least in our case study, it can be assumed that the patient sees the possibility that the treatment team shares his view that he is not mentally disturbed and that he is therefore allowed to leave the clinic again. Thus, summing up the treatment team’s situation for simplicity, there is the selection of the possibility that the patient is suffering from a serious mental disorder that justifies involuntary hospitalization. The selection of the patient who sees himself as healthy and not violent and who communicates this in this way is in the opposite position at the social level. This results in a fundamentally new situation in contrast to the state of initial complexity. A choice of action was made by both sides and communication took place. For both sides, the choice of the opposite side was of course not necessary but could have been different. This choice results in dissent, which, however, subsequently requires a new election for both sides, where, in principle, there would also be the possibility for everyone, for example, that one of the two sides would give in and a consensus would be reached, or completely different options for action would be taken that would make a consensus more likely. If we want to make an analysis of possible contradictions, we have to look more closely at the level at which the contradictions can be hidden.

Luhmann describes three dimensions of meaning: time, social and material dimensions [14]. In these three dimensions, meaning serves to present the complexity of reality in an abbreviated form in order to enable renewed actions [17].

In the time dimension, starting from the present, the past and the future are considered, whereby the present consists, on the one hand, of irreversible changes and, on the other hand, of a perceived reversibility [14]. “*For meaning systems, time is the interpretation of reality in light of the difference between past and future*” [14]. Similarly, Luhmann states that the present is experienced as “*the time span between past and future in which a change becomes irreversible*” [14]. He describes a present that lasts and stands for a reversibility that exists in systems of meaning.

In summary, in the time dimension, the constant and variable factors, which can also exist at the same time, can be viewed against each other.

“*The social dimension concerns what one at any time accepts as like oneself, as an ‘alter ego’, and it articulates the relevance of this assumption for every experience of the world and fixing of meaning*” [14]. This dimension involves the different perceptions of meaning of the various actors and the associated perception of meaning that we expect from the other or the kind of expectation expected by the other. Thus, motives and intentions in communication are attributed to one side, which simplifies the very complex social interaction. In the social dimension, therefore, reciprocal expectations or expectations of expectations play a central role, through which “*structures can be generated, against which ego and alter can be mutually oriented toward each other in their experiences and actions*” [13].

“*One can speak of the fact dimension* in relation to *all objects of meaningful intentions* (in psychic systems)” [14]. The fact dimension thus represents the thematic dimension. So, if someone communicates a topic, he again provides subsequent possibilities, which are ultimately based on ideas of process or prefigurations, in order to lend stability to communication. Essentially, a reference to the world is established via the fact dimension, which begins at the initiative of one individual and can be followed or refused by the others.

Based on these three dimensions, we can therefore better describe the situation in our fictitious department and the alienation that accompanies it.

The decision to prioritize the pharmacotherapy is made by the doctor in charge. The decision is not arbitrary, but neither is it necessary. By making the decision or better communicating the decision, the complexity of the situation is reduced and the subsequent possibilities made more clear.

When the patient comes in, there is an external reason that automatically leads to a new formation of the department’s social system.

The special feature of the social dimension [14] is that the social system is confronted with an involuntary participant; the social relationship is forced for the patient, and this results in a dissent that cannot be easily resolved in the situation described. Within the team, the treatment decisions are unilaterally attributed to the physician. The physician himself in this situation is confronted with the expectation that he is making a medical assessment and a medical decision, although perhaps all team members, as well as the patient, would see treatment options other than pharmacotherapy and would prefer them. The social system is stabilized by the given assignment in the social dimension and the associated expectations and expectations of expectations, but these create contradictions between the individual level and the social level which cannot be resolved in the situation. For the doctor, however, there seems to be a consensus that what he expects he is expected to do is the course to be followed, only the patient does not agree with this decision.

The time dimension [14] stabilizes the social system in that the reason for the patient’s admission does not change existing experiences and associated expectations but does affect the future. The team does not deviate from existing treatment schemes; the reason for the admission of the patient does not lead to an adjustment of the social system to the new situation, and there is no adjustment in connection with the other dimensions. In the time dimension, a consensus experience is also effective due to previous experiences, in that medication appears to be helpful due to the psychopathology being analogous to earlier cases.

The fact dimension [14] also shows the inconsistency of the system. Pharmacotherapy is ostensibly prescribed as an essential element of treatment by the physician. This thematically narrow specification in the fact dimension is made by the doctor because the social dimension as well as the time dimension and the associated expectations or his own expectations play roles that lead to the stabilization of the social system and to a reduction in complexity. Other team members also feel bound by this social structure. There is a fundamental consensus at the level of the social system. However, the fact that there is dissent between the personal level, or, as Luhmann calls it, the level of the psychic system [14], and the social level, both for the doctor to whom the decision is ascribed and for the other team members, conceals the consensus in the social system. The classifications that serve the consensus in the social system can therefore reduce complexity but prevent everyone involved from being able to react to their possibilities and thus pave the way for other therapy options. For the patient, it does not seem a real possibility to behave according to his own conception of possibilities; there is a superficial necessity for him to take the medication in order to make progress in therapy, which for him is associated with regaining his personal freedom. So, in the given context his autonomy is more limited, as certain actions appear necessary. In summary, we can describe an emergent social level that appears stable in itself and in which complexity is reduced by consensus, which is also understandable through the communication that takes place at the social level. However, what is also apparent is that the social level does not necessarily correspond to the possibilities of selecting the best possible treatment with regard to the patient’s autonomy. It also conceals the contradictions between the individual interaction partners.

## 5. Where Can Critique Start Here?

The essential point in the case study is the contradiction between the autonomy of the patient and the prioritization of pharmacotherapy by the healthcare professionals. On the one hand, the patient is partly allowed to maintain his autonomy in that he is not administered compulsory medication during involuntary inpatient treatment; the patient’s decision in this regard must be respected. On the other hand, the decision could lead to the patient receiving fewer or no other interdisciplinary treatment options beyond the rejected psychopharmacological treatment and to his having to spend an extended period of time receiving involuntary inpatient treatment. Therefore, a field of tension arises between respect for the autonomy of the patient—which certainly plays a role here in perception and therefore also in action—and the perception of the clinical picture, which calls for psychopharmacological therapy to the exclusion of other therapies.


*“It is therefore not only the case that in a given social context certain normative principles are (more or less accidentally) advocated which are then missed in the same context; but the corresponding social practices and institutions are themselves constituted by those norms which they simultaneously contradict.”*
[2].

If we take this quotation as a basis, possible contradictions can again be clearly stated here. The therapeutic action is based on two principles: the autonomy of the patient, an explicit norm to which all participants would certainly agree, and the “biological” decision of the doctor, which prioritizes pharmacotherapy. Other therapeutic courses of action are regarded as being not expedient—which, however, is immanently justified in the action of the team by the fact that the doctor has the professional authority here, and thus this contradiction cannot be openly discussed. The result, in our example, is that the patient remains locked up in a psychiatric ward against his will for the time being, that is, his autonomy is restricted in this area, but because of the immanent norm that psychopharmacological treatment—which the patient rejects—is effective, no further measures are adopted to restore his autonomy. This contradiction can thus be represented in the analysis of the meaning dimension.

## 6. Alienation as an Expression of Contradictory Constitutions of Meaning

In order to better embed the critique described above in a social context, it seems helpful to illuminate the issue of alienation in this overall context.

The alienation theorem, on the basis of which certain problem areas can lead to conflicts and problems in the treatment of patients, has undergone changes in recent centuries. During the 20th century, the development of this theorem of alienation was the subject of critical theory—for the so-called “Frankfurt School”, in particular. In the remainder of this study, we will refer primarily to the book about Alienation of Rahel Jaeggi [18], who developed a modern concept of alienation.

If we want to examine the structural context described above with regard to the question of what alienation means and at what level it manifests itself and how, then we have to look at the concept of alienation theoretically from the individual level and then anchor it in the structure of meaning explained above in such a way that alienation does not remain on the level of the psychic system but also inserts itself into the emergent social level.

Alienation, as mentioned in the introduction, is a concept that has a long history and exists in various forms. Rahel Jaeggi writes in her book about alienation: “*Alienation, from the perspective of the subject, is a deficient relation to the world and the self, which, my reconstruction thesis held, can be understood as a disturbed relation of appropriation: Alienation is prevented appropriation of world and self*” [19].

Accordingly, alienation, or the suspension of alienation, should not be understood as an achieved state but rather as a continuous process—and the self is only formed through this process. Jaeggi also emphasizes that a non-alienated self is not a self that rests within itself but is always a process that is mediated in part through the appropriation of the world. “*Alienation means the stoppage of experiential processes. And: alienated is he who cannot relate to his presuppositions, who cannot appropriate his presuppositions*” [19].

Jaeggi later explicates the social contexts more precisely [20] by emphasizing that alienation is not to be found only on the side of the subject; there are always counterparts in the external world—be they institutions or social relations—which are also the target of critique. Social practice, according to Jaeggi, is simultaneously “*alienated and alienating*” [21], and it “*produces alienation and is itself the product of alienation*” [20].

As she indicates in her book *Critique of Forms of Life*, social practices are usually sequential acts that are performed not just once but repeatedly. “*Practices, then, are habitual, rule-governed, socially significant complexes of interlocking actions that are enabling in character and by which purposes are pursued*” [21]. A social practice is also not to be perceived as a single entity, as it offers a social context and multi-directional referential contexts: “*Individual social practices thus have antecedents and connections to other practices. Practices are thus networked with manifold other practices and settings, in the context of which their function and specific meaning are first gained*” [21].

Thompson [22] describes alienation more specifically as “atrophied moral cognition,” by which he means that humans can no longer distinguish between right and wrong in their actions on their own. The ability to make this distinction is, what he calls, moral cognition. According to Thompson, people grow up in a value system that determines both their thinking and their behavior to the extent that decisions about right and wrong are also transmitted to the outside. In a state of alienation, people can no longer reflect on this distinction themselves, as it depends on external value systems, which, depending on the imprinting, are represented by institutions or authorities.


*“I am alienated, in other words, in this sense, when I am no longer able to ground the reasons for my actions, my beliefs, and my practices and commitments but instead rely on an already pre-formed set of reasons, rationales, and values that make the closed system of my social world legitimate to me, knowable, predictable, and so on. Alienation, on this view, is a deeper account of the moral-cognitive and rational processes that are responsible for my autonomy.”*
[22].

If we apply this explanation to our example, we locate a fundamental problem in the discussion after the patient has left. Due to the perspective strongly focused on psychopharmacology, which in this case is mainly shaped by the doctor, it is difficult for the individual team members who also want to consider other treatment options according to their competences and training to use these options. This means that the possibilities for action, which are also given in the social space of the ward, are limited, which is reflected in the meaning dimension, especially the allocation of meaning externally. In other words, decisions are assigned externally, so that right and wrong seem to be decided, above all, by the structure of predetermined values and evaluations prevailing on a clinical ward.

Georg Rilinger critically notes in this regard that the philosophical concept of alienation is difficult to adapt to empirical material because it is formed purely philosophically [23]. Rilinger proposes to solve this problem by subjecting the conceptualization of the theory to the circular process of immanent critique as well, thereby developing the concept of alienation on the basis of “discursively negotiated interpretations” [23], which would achieve a parallelism of application as well as theoretical development. In the following, we will pursue an approach that counters this criticism and enables applicability and further development at a theoretical level.

## 7. Meaning and Alienation

An essential step to relate the problem to acute psychiatry and thus put it into practice is still the coupling of the two theories described above: first, the double-contingency structure of the social and, second, the theory of alienation described above.

To further illustrate the connection, let us recall our case study again. There we see various alleged guidelines—legal, social and also medical–scientific—which ultimately give the healthcare professionals the impression that, on the one hand, there is one prioritized way of treating the patient, namely, with medication, while, on the other hand, at least for the time being, coercion appears to be valid and is also legally justified in keeping the patient in the clinic against his will. The patient thus encounters a rigid system when he is admitted as an inpatient. For the treatment team, the patient is the external cause, and each individual is unable to adequately respond to their own possibilities in the treatment of the patient, which certainly go far beyond the mere administration of medication. This creates alienation on the therapeutic side as well as on the side of the patient; all, with their individual needs and problems, encounter rigid structures.

The structuring of our social environment occurs through attributions in the various dimensions of meaning, which are associated with each other, with different weightings depending on the context of life in question [14]. The complexity present in double contingency is reduced by the fact that communication takes place as action and thus as selection, such that meaning can be attributed to it [14].

If we consider alienation as a phenomenon that manifests itself in the association of all three levels of meaning, we can pinpoint a connection between the two theories. If we begin from a social practice [21], as described above, patterns of alienation can be identified in these dimensions of meaning.

In order to be able to present this connection, we will present this possibility of alienation in all three dimensions of meaning.

In the time dimension, a social practice serves as something constant, relatively stable, which receives its stability through expectations and expectations of expectations [9,14]. This constancy, however, affects the relationship to the world, so that it is difficult for a sometimes-necessary appropriation of the world and the self to take place in the process of a social practice. Selection possibilities that can actually be seen as variable seem unlikely or impossible, as certain selections are made out of the constancy of the practice. In the time dimension, the inconsistency of the system becomes clear again, since individual team members cannot respond to individual needs due to the constancy of the treatment process. The doctor, to take him as an example, may have other treatment options, just like the other team members. The expectations and expected expectations appear constant over time, so that an adapted and individual reaction to the patient is not possible.

In the social dimension, the potential for alienation in social practice lies in an assignment of motives for action being left to one side, so that an individual may not see themself as an actor [14]. The person’s motives cannot be realized: they are either attributed to others—that is, hierarchically higher persons—or abstracted into generality—“one always does it this way.” In our example, the contradiction in the social dimension is shown very well in that the decision to prioritize pharmacotherapy is unilaterally attributed to the doctor and, contrary to his own possibilities, he expects that the other team members expect him to make the decision for the prioritization of pharmacotherapy.

The fact dimension [14], that is, the topic at which communication is directed, has potential for alienation insofar as social practices or even institutions in differentiated (sub-)systems are rigid in their respective fact dimensions and can only be acted upon to a limited extent in response to external occasions. Alternatively, one cannot use the referrals in the fact dimension as one wishes; rather, the possibilities of connection are fixed in the environment, as a result of which one can no longer relate to one’s preconditions, as described above.

In order to summarize the text, from our point of view, a system must not only be criticized for contradictions between norms and the system’s social practices. Central to this is the problematic difference between the individual level and the social level, which can be explained using Luhmann’s dimensions of meaning. In this way, rigidities in a social system that are strongly based on expectations and expectations of expectations can be criticized and the system can be transformed and allowed to react more flexibly. The aim is to reduce alienation for the patient as well as for staff by reducing these contradictions. Especially in acute psychiatry and in the treatment of involuntarily admitted patients, it is important to make these contradictions clear and to remove immanent obstacles to good treatment.

## 8. Discussion

In summary, the aim of this work is to show that existing modes of action, patterns of action or social practices in acute psychiatry are not always to be taken for granted; certain immanent contradictions and conflicts should be made accessible through an analysis of the dimensions of meaning. Another central aspect of this work is that it is not primarily about criticizing the entire psychiatric system. Rather, the aim is to better understand and analyze conflicts and contradictions in the social system at the level of the ward or perhaps of the clinic, that is, “as analysis criticism […] and as criticism analysis” [2].

In principle, a more detailed examination of this issue, also in relation to psychiatry as an organizational system, would have to be carried out in greater detail, which is not possible here. There are already studies devoted to the immanent critique in the psychiatric field—in particular, on the basis of qualitative research, especially by means of interviews—but these mainly illuminate changes in the larger social or economic frameworks [24]. This organizational problem becomes somewhat clearer, for example, with Martin Herberg:


*“Every communicative event generates surpluses of possibilities from which something suitable is selected in the next step. The decision about what is suitable and what is not is made on the basis of special, internal organizational decision premises, which in their totality result in the structure of an organization.”*
 [25].

In order to be able to examine the fine structure of a course of psychiatric treatment, the possibility of analyzing the meaning and thus the possibilities of action that arise in the social system of an acute psychiatric ward seems to us to be purposeful, especially if one wants to recognize alienation potentials.

In the three dimensions described by Luhmann, an established externalization of references to action can also be made accessible, in that the constituents can be questioned so that they can be clearly perceived as more changeable within the system.

In the following text, the above-mentioned problems are placed in a context wherein certain perspectives—of course, only exemplary perspectives—are illuminated. In particular, the contradiction between autonomy and coercion, whereby coercion also encompasses being held in an institution against one’s will, repeatedly leads to wide-ranging discussions in psychiatry and ethics, as well as in legislation.

Giovanni Maio has contributed to the literature by discussing the contradiction that arises when a person suffering from a serious mental illness refuses help that they need in order to be able to live autonomously again. In such a case, if the help refused by the patient were not provided, “*one would be invoking autonomy and at the same time acting against it*” [26]. Maio clearly refers to the fact that in the case of a severe mental illness, one cannot speak of autonomous will formation. In principle, this view appears to be purposeful, but one must consider the question of the severity of mental illness and the evaluation of autonomous will formation. They are regarded as constitutive of this approach and can only be objectified to a limited extent, and therefore can only be applied as a binding external standard to a limited extent; this view can also lead to internal contradictions, which can then be fundamentally criticized immanently, ushering in a transformation. The method described above is therefore not in contradiction with this attitude; rather, it is important to consider that robust ethical foundations for clinical work can achieve a transformation at the executive level, thereby negating unnecessary restrictions and regulations.

In the case of a one-sided fixation on psychopharmacological treatment, high legal hurdles for such treatment against a patient’s will often be the reason for therapy discontinuation—if there is no immediate danger—on the part of the treatment provider, despite the knowledge that in due course of time after discharge the danger may recur. “*Behind this practice, in addition to purely pragmatic reasons, lie partly unresolved conflicts pertaining to different concepts of illness, paternalistically oriented medicine and psychiatry, and autonomy rights of the individual, as well as legislation and psychiatry.*” [27].

Additionally, according to Botlender and Juckel [27], personal and situational factors often play a joint role. So, apart from factors such as poor training or lack of experience, personal life circumstances can also influence an individual’s behavior and contribute to increased aggression and violence.

A Norwegian study [28] on the normative attitudes of staff showed, for example, that attitudes toward coercion may depend in part on hierarchical levels or professional groups, as well as on the specialization of departments. In particular, senior staff members with decision-making responsibilities were less likely to view coercion as therapeutic, and psychologists were generally most critical of coercion. One possible explanation given was that psychologists were more likely to take a relationship-based approach to the patient, whereas psychiatrists were more likely to think in medical terms of diseases to be treated and risks to be avoided. The authors concluded that staff members do not often feel a moral doubt regarding the use of coercion and that staff who were more involved in the use of coercion were also more likely to regard coercion as caring.

In our case report, an ethical issue we want to focus on is shared decision making. Shared decision making means that treatment options are explained to the patient as clearly as possible and that the decision about the course of treatment is made not only by the clinician but with the involvement of the patient. Studies examining the role of shared decision making show that even involuntarily admitted patients had better experiences when they could participate in decision making [29]. In a qualitative study, Giacco et al. [30] identified different barriers and facilitators in shared decision making. Possible barriers are difficulties in communication, such as the difficulties patients may have in communicating their problems, the use of terminology by clinicians that patients do not understand and a lack of communication techniques on the part of clinicians. Another barrier identified is the noisy and busy ward environment and “*immovable dynamics and policies of the ward*”. As possible facilitators, the authors identified starting shared decision making as soon as possible, before patients have too many negative experiences; the involvement of the whole clinical team; and positive relationships between patients and staff members. The involvement of caregivers was found to have advantages as well as disadvantages. These factors lead to problematic courses of treatment. Van Kranenburg et al. [31] found that in long-term treatment, besides the experience of improvement, it is also important for the quality of life of patients that they have the feeling of being taken seriously. The importance of shared decision making for treatment satisfaction and less decisional conflict in general as outcome parameters was shown in a meta-analysis by Shay et al. [29]. If we discuss these factors in relation to our theory, we can transfer these points to the social structure in our case. Due to the prioritization of pharmacotherapy, the patient feels misunderstood and cannot explain his problems, the seemingly immovable dynamics of the ward constituting a social system that conceals contradictions and where there is a lack of involvement of the whole team due to their ascribing competency regarding the treatment decision to the physician. Even if the patient could not leave the clinic, the quality of life could be better if the patient were taken seriously. Methodologically, we think that our approach could lead to improved shared decision making by transforming the system, making it less contradictory, so that every team member is part of the decision making and can bring his own possibilities and skills into the treatment. There is certainly the essential limitation with regard to therapeutic decision making that, due to the severity of symptoms, certain medical measures to improve a patient’s condition may be necessary. This often affects the field of psychopharmacology. Patients may also be rejected if the severity of symptoms does not allow insight into the need for drug treatment and alternative treatments are not promising.

In modern psychotherapy, in addition to a positive relationship, a constructive approach to the psychotic symptoms of the patient is in the foreground. In recent psychotherapeutic research, good results have been obtained when the patient has not been fixated on the illness value of their symptoms, the work taking place in the patient’s belief system instead [32]. Therapy goals should therefore be formulated with reference to the patient’s beliefs and should not contradict the patient’s psychotic content, the aim being to treat the patient on an evidence-based basis [33]. Overall, this has made it more possible to formulate viable treatment goals for most patients [34].

In order to carry out such a psychotherapeutic treatment in an acute ward, we believe that, in addition to the implementation of the method, it is just as important that the social system of the ward react flexibly as cases arise, i.e., when patients are admitted. In an alienating system such as we have described, the problem becomes clear, since the system, due to the social structures predetermined by the expectations and expectations of expectations, does not adapt itself as well as if it were acceptable to the patient with its fundamentally available resources; practically, it cannot react.

Possible benefits of applying our theory in practice would be the empowerment of patients and all team members by the uncovering of hidden contradictions and their criticism via immanent critique. So, taking part actively in the decision-making process may lead to greater satisfaction among nurses [35]. Reducing alienation should also help to improve the self-efficacy of the team-members. In a Japanese study [36], for example, the authors identified factors that improve self-efficacy, i.e., positive reactions in patients and the ability to positively change the nurse–patient relationship, and factors associated with decreased self-efficacy, such as the loss of roles for nurses and uncertainty in psychiatric nursing.

The limitations of this method and the risks involved relate to the legal roles of the team members, especially the role of the physician. Especially in emergency situations, the physician is responsible for decisions, in particular, decisions about whether a medication is to be applied against the will of a patient or whether seclusion is necessary. In these cases, transformation is limited, because the physician often has the whole responsibility and decisions have to be made quickly. There are also legal limitations to what is allowed for each group of healthcare professionals in these situations [37]. We see a further limitation of the study in the fact that the theory is not substantiated by empirical study data but is rather based on clinical experiences that were not systematically collected. The article contains not only an objective description of the theory and discussion of it in the context of acute psychiatric treatment, but also the personal experiences and opinions of the authors.

The problems that this study aims at resolving are partly found in openly stated conflicts but often also manifest as tensions and conflicts within teams, finding expression as ethical–moral doubts or the alienation of individuals from their own work. Basically, a meaning analysis at the individual and social levels seems useful in order to be able to reveal the contradictions which often create difficulties in the implementation of therapies tailored to patients’ needs and which comply with current treatment guidelines. These contradictions are found not only in the patient–therapist relationship, but also in the relationships among the therapists themselves, as well as in the organizational structure.

The theory elaborated here makes a more concrete application of alienation theorems in social systems possible because it seems to us to be an essential point that immanent contradictions are an essential part of conflicts and tensions that lead to the alienation of all those involved in the therapeutic process. In addition, it should also serve as a basis for initiating transformation processes and reducing alienation through immanent critique. Finally, it seems important to us to mention that we do not consider a psychiatric ward as a closed social system that can be transformed only by immanent critique.

In order to open up a possible application perspective, there are already various possibilities in psychiatry. From our point of view, supervision seems to be valuable, apart from the therapeutic problems, in addressing expectations, but also expectations of expectations, mutually, in order to reveal contradictions independently of the professional discussion about the right therapy in order to reduce the alienation of the individual team members and to bring about an improvement in the social system at a structural level. This is also intended to ensure that each member of the therapeutic team can contribute better to shaping the relationship. In order to reduce the alienation of the patient, it seems necessary that, as far as possible, therapy decisions not be made under time pressure and one-sidedly by the doctor; rather, the patient should feel that they are being taken seriously, and this can be achieved by dismantling alienating social structures through system transformation.

## 9. Conclusions

In summary, our work presents a method for immanent critique in acute psychiatric wards. Social subsystems, such as psychiatric acute care units, can be alienating for both patients and staff, ultimately affecting the quality of care. Our approach uses Niklas Luhmann’s concept of meaning, which is also linked to the importance of the expectations that stabilize a system. In order to go beyond this analysis, where the concept of meaning has been used to enable critique, we combined this method with the concept of alienation as a lack of appropriation of the world to arrive at a method that can represent system-immanent contradictions between the social and individual levels and thus make seemingly rigid systems transformable.

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
