# Peer review of "Meaning Analysis and Alienation: A Method of Immanent Critique in Acute Psychiatry"

_ijerph, 2022, doi:10.3390/ijerph192316194_

Round 1

Reviewer 1 Report

Congratulations. This is an article that addresses a necessary topic with a very appropriate approach.

-The title, abstract and keywords are correct.

-The structure of the report is coherent and scientifically consistent.

-It is necessary to review sections 6. Alienation as an Expression of Contradictory Constitution of Meaning and 7. Meaning and Alienation, since its presentation can be improved. Perhaps by ordering the existing information a less chaotic aspect is obtained.

-There are many paragraphs throughout the text without any bibliographical reference, more references should be added to the text.

-The style of bibliographical references of this journal should be consulted. Currently, the style is not correct.

Author Response

Dear reviewer,

thank you for the important and helpful comments. In accordance with your first revision note, we have rearranged Sections 6 and 7 and hope that they are now clearer and more readable.

In the longer sections without quotations, we have now added some bibliographical references in order to establish a link to the original literature sources in these sections as well.

We have also adjusted the bibliography.

Based on the comments of the other reviewer, we expanded the case study to include methodological notes and limitations. In addition, we have elaborated on the problem of non-consistency that cannot be achieved and added the section on shared decision-making. We also removed a paragraph from the discussion. We hope that with these revisions we have been able to adapt the text to your points. If we were not able to do this, we would be grateful for further suggestions for improvement.

With kind regards

The authors

Reviewer 2 Report

Thank you for inviting me to review this article. It is a thought-provoking and interesting piece of analysis which I enjoyed reading. The imminent critique is developed and argued well, and some interesting proposals are made in the discussion. Overall I think this has significant merit, though I do have some reservations about the current version.

The main problem I had with the article was the use of a single hypothetical case-study to inform the entire imminent critique. There are good reasons not to utilise real/ empirical case studies in this sort of sensitive situation, of course, but I don't think this hypothetical one is actually necessary to develop the critique. You state that this was derived from clinical experience in a footnote, but you could make the same points, perhaps more effectively, if you draw upon your experiences, about, for example, the tensions in pharmacotherapy at individual and social levels, or the more general sense of alienation which occurs in psychiatric wards. Additionally, the case study is somewhat under-developed in some key areas. For example, at the end of the case study it is briefly mentioned that treatment options had not been exhausted, and then throughout the critique, the possibility of nonpharmacological treatments is alluded to. I agree with this in a general sense, but what exactly do you have in mind here? What other options would be suitable for a patient with this diagnosis? So, while this is a rich article in many ways, I did feel the centrepoint was somewhat flimsy in comparison to the wider critique in which you engage. 

Alternatively, I'd need to see a much clearer explanation of how the case study was developed and analysed, and a discussion of the methodological limitations of using this approach.

Also, I did wonder to what extent a meaningful transformation of the psychiatric clinic in a less alienated direction is even possible. This is a debate that goes back to the work of R.D. Laing, and even before that. Setting aside the historical instances where 'patients' have been interred in a ward against their will for no sound reason, aren't matters different where someone poses a direct threat to themselves and/ or others? You acknowledge this on p. 4, but I don't think you quite follow the argument to its conclusion: "Moreover, in a critical area such as 155 acute psychiatry, the absence of contradictions seems difficult to achieve because patients often move in a context that inevitably means a contradiction to their own ideas and needs." Is it not the case that alienation exists outside, and prior to the clinic, and that the alienation of the clinic is in a sense secondary to this?

There is also the issue of shared decision-making you raise in the discussion. Again, there are limitations to this if the individual is not capable, so it perhaps another tension which is difficult to resolve in psychiatry.

Finally, although you are talking about the 'environment' of the psychiatric ward in a general way, linking the analysis to the notion of environment (in the sense of a specialised space separate from broader society) could make the relevance to the journal more clear. 

Minor comments:

Typo on p. 2: "expectations what leads to alienation" should be "expectations that lead to alienation"

Bottom of p. 3, the system "does not become too rigid and open to criticism" - should that be "closed off to criticism"?

"Hoff highlights the challenge of the inclusion of at least three levels: social, psycho- 586 pathological, and biological. However, “abusive reduction (reductionism in the negative sense) 587 and, in the worst case, dogmatic ossification can be achieved at any of these levels and with any 588 scientific method.” (22) In conclusion, he refers to “the inextricable link between clinical issues 589 and philosophical and ethical aspects” (22) and, consequently, that “psychiatry has the authority 590 and the obligation to insist on keeping the debate about basic anthropological assumptions open.” 591 (22)" - I don't see how this adds anything to the discuission - suggest you remove this bit.

Author Response

Dear reviewer,

thank you for the important and helpful comments.

We can understand your criticism of the constructed case study very well, we have decided to prefer the alternative you suggested in the hope that we could improve this part of the work. We have expanded the case study to include explanations regarding the development and limitations and once again clarified the focus on psychopharmacology. In the case study itself, we have added some clarifying elements, in particular possibilities for non-pharmacological interventions have been presented more clearly.

As to your next comment on whether transformation in a less alienating direction is possible, we think, also from our experience, that it is possible. We have included an explanation of this at the end of Section 2. We are absolutely of your opinion that the alienation for the patients is certainly primarily outside the clinic and that the alienation in the clinic follows inevitably, i.e. is secondary. The same applies to many factors that have an external impact on the subsystem of the psychiatric acute care unit. However, we see the basic potential of the treatment in the fact that a transformable and therefore also learnable and more flexible system on a ward improves the success of the therapy and the patient is correspondingly less alienated from the world outside the clinic. For this it is also important that the treatment team as a whole also counteracts his alienation and thus also the alienating effect. A freedom from contradiction is certainly not achievable, rather we see it as a task to work on the contradictions and to keep the system transformable. We hope that you can understand this point in this way. We have added the limitation that shared decision-making is not possible in serious circumstances in which the patient cannot agree to urgently needed treatment due to the severity of his or her disorder. In the introduction we have made the reference to the wider environment in order to make the reference to the journal better.

Thank you for the other minor comments, we have adjusted the text accordingly.

Following the other reviewer's comments, we have restructured Sections 6 and 7 to make the text more organized and readable. We have also added bibliographic references to some sections.

We hope that in our revision we have been able to improve our text according to your comments. Of course, we are open to further suggestions for improvement and will continue to implement them.

Kind regards

the authors

Round 2

Reviewer 2 Report

Thank you for inviting me to re-review the manuscript. Overall the changes made have significantly improved the work and I think it is now of a good enough standard for publication. I still feel the hypothetical case study is not necessarily the strongest basis for developing the theory the authors propose - however, this is balanced out by the depth of the philosophical critique offered, and additionally the authors have expanded and clarified the case study well.